# Melatonin Receptor Expression in Primary Uveal Melanoma

**DOI:** 10.3390/ijms25168711

**Published:** 2024-08-09

**Authors:** Anna Hagström, Ruba Kal Omar, Hans Witzenhausen, Emma Lardner, Oran Abdiu, Gustav Stålhammar

**Affiliations:** 1Department of Clinical Neuroscience, Division of Eye and Vision, Karolinska Institutet, 171 77 Stockholm, Sweden; ruba.kalomar@regionhalland.se (R.K.O.); hans.witzenhausen@stud.ki.se (H.W.);; 2St. Erik Eye Hospital, 17164 Stockholm, Sweden; emma.lardner@regionstockholm.se; 3Ögonspecialisterna Farsta, 12347 Stockholm, Sweden; oran@ogonfarsta.se

**Keywords:** adjuvant treatment, immunohistochemistry, melatonin, metastasis, prognostic factors, receptor expression, uveal melanoma

## Abstract

Melatonin, noted for its anti-cancer properties in various malignancies, including cutaneous melanoma, shows promise in Uveal melanoma (UM) treatment. This study aimed to evaluate melatonin receptor expression in primary UM and its association with UM-related mortality and prognostic factors. Immunohistochemical analysis of 47 primary UM tissues showed low expression of melatonin receptor 1A (MTNR1A) and melatonin receptor 1B (MTNR1B), with MTNR1A significantly higher in patients who succumbed to UM. Analysis of TCGA data from 80 UM patients revealed RNA expression for MTNR1A, retinoic acid-related orphan receptor alpha (RORα), and N-ribosyldihydronicotinamide:quinone oxidoreductase (NQO2), but not MTNR1B or G protein-coupled receptor 50 (GPR50). Higher MTNR1A RNA levels were observed in patients with a BRCA1 Associated Protein 1 (BAP1) mutation, and higher NQO2 RNA levels were noted in patients with the epithelioid tumor cell type. However, Kaplan–Meier analysis did not show distinct survival probabilities based on receptor expression. This study concludes that UM clinical samples express melatonin receptors, suggesting a potential mechanism for melatonin’s anti-cancer effects. Despite finding higher MTNR1A expression in patients who died of UM, no survival differences were observed.

## 1. Introduction

### 1.1. Uveal Melanoma

Uveal melanoma (UM) is the most prevalent primary intraocular malignancy in adults [1]. Within 15 years following diagnosis, between a third and half of all patients succumb to metastatic disease, irrespective of whether they undergo primary treatment with plaque brachytherapy or surgical enucleation [2,3,4]. This may be explained by the early spread of small tumor deposits, known as micrometastases, from the eye to distant organs, primarily the liver [5]. Once these clusters of tumor cells evolve into larger, radiologically detectable masses, median survival is approximately one year [6]. Unlike cutaneous melanoma, treatments involving immune checkpoint and v-Raf murine sarcoma viral oncogene homolog B1 (BRAF) inhibition have shown limited efficacy in extending survival for those with metastasized uveal melanoma [7].

Since macrometastases are uncommon at diagnosis, predicting the prognosis for individual patients relies on clinical and histological aspects of the primary tumor [8]. These factors include tumor location, size, and configuration as well as tumor cell type and the presence of genetic mutations or chromosomal abnormalities [9]. Primary uveal melanomas consist of four cell types; normal, spindle A, spindle B, and epithelioid cells where the last-mentioned has lower expression levels of the BReast CAncer gene associated protein (BAP-1) [10]. The BAP-1 gene is located on chromosome 3p21.1 and is involved in epigenetic modulation of chromatin, DNA- and cell repair as well as tumor growth suppression [11,12,13]. It is one of the genes mutated in uveal melanoma, often occurring in later stages of the disease with low nuclear BAP immunoreactivity being significantly associated with a higher incidence of metastasis [14]. Inactivating mutations of BAP-1 have been associated with monosomy 3 where protein absence requires biallelic changes through the loss of a copy on chromosome 3 and mutations of the remaining copy [15]. Monosomy 3 is related to poor prognosis of uveal melanoma regarding survival after treatment, suggesting a potential tumor suppressing role of chromosome 3 [16]. Furthermore, there appears to be a correlation between BAP-1 inactivation and the presence of the epithelioid cell type in primary tumors where epithelioid-mixed cell types are associated with worse prognosis [15].

### 1.2. Melatonin

Melatonin is a hormone that has been found to have therapeutic benefits in patients with cancer [17]. The indoleamine is primarily produced in the pineal gland and secreted into the bloodstream and cerebrospinal fluid in the evening [18,19]. In addition to regulating the circadian rhythm, melatonin impacts several other bodily functions, has been described as a potent free radical scavenger and appears to aid in DNA repair [20,21]. Moreover, melatonin contributes to immune system enhancement through promotion of T-helper cells and regulation of the maturation of immune cells including T-, NK-, and B-cells [22]. When used as an adjuvant treatment for various types of cancer in previous studies, melatonin has been associated with inhibited tumor growth, increased one-year survival, and few side effects [23,24]. The hormone has been shown to have low toxicity on its own while reducing the toxicity of chemotherapy and radiotherapy treatments [24,25,26].

### 1.3. Melatonin Receptors

#### 1.3.1. Melatonin Receptor Type 1A and Melatonin Receptor Type 1B

Melatonin impacts various processes in the human body primarily through two membrane receptors; melatonin receptor type 1A (MTNR1A) and melatonin receptor type 1B (MTNR1B) [27]. Both receptors are G-coupled receptor proteins widely expressed both centrally and peripherally throughout the human body and play a role in melatonin’s impact on physiological systems such as circadian rhythm, neurodevelopment, blood glucose regulation and the cardiovascular system [28,29]. In a study published in 2000, Roberts et al. identified the MTNR1B subtype, but not the MTNR1A subtype, in reverse-transcribed RNA obtained from normal uveal melanocytes as well as melanoma cell lines [30]. In the study, receptor agonists for both MTNR1A and MTNR1B as well as melatonin itself inhibited the growth of uveal melanoma cells at physiological concentrations, thereby suggesting a receptor-mediated mechanism for the inhibition of tumor cell growth [30]. MTNR1B appeared to be expressed to a larger extent in the cancerous cell lines compared to normal uveal melanocytes, though this observation was not specifically verified [30].

#### 1.3.2. N-Ribosyldihydronicotinamide:Quinone Oxidoreductase 2

N-ribosyldihydronicotinamide:quinone oxidoreductase 2 (NQO2) presents a melatonin binding site known as MT3 and carries out two-electron reductions of, primarily, quinones [31,32]. Despite previously being considered a detoxifying enzyme, some studies suggest that NQO2 activation leads to the production of reactive oxygen species (ROS) and the enzyme has been found to be increased in some cancer cell lines [33,34]. Of note, the affinity of NQO2 to melatonin seems to be in the nanomolar range, however, melatonin appears to inhibit the enzyme within the micromolar range, where concentrations above 1 μM are considered pharmacological [32].

#### 1.3.3. Retinoic Acid-Related Orphan Receptor Alpha

The retinoic acid-related orphan receptor alpha (RORα) is a nuclear receptor within the ROR family [35]. RORα has been found to play a role in the immune system by contributing to Th17 development and the generation of innate lymphoid cells [36,37]. It may also aid in the stabilization and transcription of p53 [38]. Several studies suggest that RORα expression is down-regulated during tumor development and progression, while exogenous RORα inhibits cell proliferation and tumor growth in colorectal, prostate and breast cancer [39,40,41]. Previous immunocytochemistry studies on human melanoma samples have demonstrated an association between increased melanin pigmentation and reduced expression of RORα [42]. Interestingly, vitamin D hydroxyderivatives can act as inverse agonists to the receptor and have been found to possess anti-melanoma properties and protective effects against oxidative stress and DNA damage as in the case of melatonin [43]. Whether a correlation between melatonin and the ROR family exists has been debated, however, one study, using crystallography and molecular modeling, indicated that melatonin is unlikely to function as an ROR ligand and is therefore not a melatonin receptor [44,45]. Despite this, specific intermediate steps enabling melatonin to indirectly regulate ROR expression and function have been confirmed [44].

#### 1.3.4. G Protein-Coupled Receptor 50

GPR50 is a G protein-coupled receptor with the ability to heterodimerize with both MTNR1A and MTNR1B [46]. GPR50 does not seem to modify the function of MTNR1A but rather leads to an inhibition of the functional response of the receptor to stimulation by melatonin [46].

### 1.4. Aim of the Study

Past studies have demonstrated that melatonin inhibits growth of cultured human uveal melanoma cells, however the mechanism by which this occurs is not fully understood [47,48]. If melatonin influences the progression of uveal melanoma and slows the onset of macrometastases, one potential mechanism could be through binding melatonin receptors present in primary uveal melanoma cells. This study aimed to explore expression levels for melatonin receptors in primary uveal melanoma and investigate a potential correlation between receptor expression and the risk for uveal melanoma related mortality as well as other prognostic factors such as *BAP1* expression, cell type, and monosomy 3. We examined the expression of melatonin receptors in uveal melanoma tumors from two separate cohorts and identified the presence of four receptors including the two main melatonin receptors MTNR1A and MTNR1B as well as RORα and NQO2. Higher levels of MTNR1A were found in patients who died of UM, however, Kaplan–Meier analysis showed no difference in the survival curves as they related to receptor expression in either cohort.

## 2. Results

### 2.1. Descriptive Statistics

Analyses of the distribution of melatonin receptor expression between males and females revealed no significant differences in immunohistochemical expression of MTNR1A or MTNR1B in neither the cytoplasm nor nuclei (Mann-Whitney U *p* > 0.20) in the St. Erik cohort. Similarly, there were no significant differences in the expression levels of RORα, NQO2, or GPR50 RNA between the sexes (Mann-Whitney U *p* > 0.68) in the TCGA cohort.

### 2.2. Melatonin Receptor Type 1A

MTNR1A was expressed in uveal melanoma tumors in both the St. Erik and the TCGA cohorts (Figure 1). However, no significant difference was seen in survival probability between patients in the TCGA cohort with MTNR1A RNA levels below vs. above the median TPM value or below vs. above 1 TPM. Similarly, no significant difference was seen in survival probability between patients in the St. Erik cohort with MTNR1A optical density (OD) values below vs. above the median. In the St. Erik cohort, slides were missing for one of the tumors resulting in a total of 46 tumors analyzed. As illustrated in Figure 2, the median MTNR1A OD was significantly higher in the cytoplasm compared to the nuclei (*p* < 0.001). There was also a higher MTNR1A OD in the cytoplasm of those who died from uveal melanoma compared to those who did not (Holm-Bonferroni corrected *p* = 0.004). No significant difference was seen between UM related death and nuclear MTNR1A OD (Table 1). In the TCGA cohort, MTNR1A RNA levels were significantly lower in tumors with BAP1 mutations (Holm-Bonferroni corrected *p* = 0.005) (Table 2C) while there was no correlation between MTNR1A RNA and cell type or Monosomy 3 (Table 2A,B).

### 2.3. Melatonin Receptor Type 1B

No tumors from the TCGA data expressed MTNR1B (Table 2). In the St. Erik cohort of 47 patients, expression levels were analyzed in the nuclei and cytoplasm of 45 tumor samples as the slides immunohistochemically stained for MTNR1B were missing for two patients. As in the case of MTNR1A, the median MTNR1B optical density (OD) was significantly higher in the cytoplasm compared to the nuclei (*p* < 0.001) as described in Figure 2. No significant difference was seen between receptor expression and survival probability. Similarly, no correlation was observed between UM related death and OD levels in the cytoplasm or nucleus (Figure 3, Table 1).

### 2.4. N-Ribosyldihydronicotinamide:Quinone Oxidoreductase 2

The median NQO2 RNA expression levels for the 80 tumors in the TCGA data was 36.37 TPM. There was no correlation between receptor expression and survival probability (Figure A1). NQO2 RNA expression was higher in patients with epithelioid tumors compared to those with either spindle or mixed cell types (*p* = 0.01); however, after the Holm–Bonferroni correction, the result was no longer significant (corrected *p* = 0.05), as described in Table 2A. Similarly, when comparing NQO2 RNA expression levels across the three separate cell types, i.e., epithelioid, spindle and mixed, using the Kruskal–Wallis test, the original *p* value was significant (*p* = 0.0468) while the Holm–Bonferroni corrected *p* value was not (corrected *p* = 0.234). No other correlation was noted for other prognostic factors (Table 2B,C).

### 2.5. Retinoic Acid-Related Orphan Receptor Alpha

The median RNA expression level of RORα among the 80 tumors in the TCGA cohort was 0.66 TPM. Twenty-five patients had an RNA expression level equal to or above 1 TPM while 55 patients had an expression level below 1 TPM. No correlation between expression and survival probability or prognostic factors was noted (Figure A2, Table 2).

### 2.6. G Protein-Coupled Receptor 50

No tumors from the TCGA data expressed GPR50, i.e., the median expression level was 0 TPM with no tumors having a mean RNA expression level over 1 TPM (Table 2, Figure A3). There were therefore no observed correlations to UM survival or any of the included prognostic factors).

## 3. Materials and Methods

### 3.1. Patients and Samples

In order to determine expression levels of melatonin receptors in uveal melanoma, data were collected from both The Cancer Genome Atlas (TCGA) as well as from immunohistochemically stained tissues from the ocular pathology archive at St. Erik Eye Hospital in Stockholm, Sweden (Table 3). Within the Swedish population of 10 million, all uveal melanomas are diagnosed at this hospital, allowing access to extensive clinical and survival data with minimal loss to follow-up. St. Erik Eye Hospital is also the nation’s only laboratory for ocular pathology, with an extensive archive of almost all eyes and periocular tissues that have been surgically removed and examined since the 1960’s.

Fourty-eight eyes with uveal melanoma, enucleated between the years 2000–2008, were collected from the archives of the St. Erik Ophthalmic Pathology Laboratory for this study after obtaining approval from the Swedish Ethical Review Authority (year 2024, reference number 2024-00295-02). One eye was later excluded as medical records revealed the tumor originated from the conjunctiva. Formalin-fixed paraffin-embedded tissue from the enucleated eyes were sectioned and stained on a BOND III IHC stainer by biomedical analysts using mouse monoclonal antibodies to detect melatonin receptor 1A (NBP3-03633, Novus Biologicals, Centennial, CO, USA) and melatonin receptor 1B (NLS932, Novus Biologicals, Centennial, CO, USA). The hematoxylin-stained tissue sections were then digitally scanned using a Grundium Ocus 40 digital slide scanner (Grundium Oy, Tampere, Finland). Mean levels of immunohistochemical expression of MTNR1A and MTNR1B across all tumor cells, as determined by their staining intensity, were quantified using optical density (OD) in bioimage analysis (MTNR1A OD, MTNR1B OD). This was done using the program QuPath Bioimage analysis v. 0.4.1 m4 as illustrated in Figure 4 [49]. In QuPath, mean DAB (3,3′-diaminobenzidine) staining intensities were calculated from the sectioned and stained primary uveal melanoma tissue where DAB indicated the presence of the melatonin receptors. Cells which only revealed the background hematoxylin staining were considered negative. Calibration was performed by manually selecting a positive and negative cell. After calibration, three circular sections, 500 μm in diameter, were selected for analysis using the positive cell detection feature in QuPath which can further separate the nucleus from the cytoplasm in each cell. From this analysis, the mean staining intensity in both the nucleus and cytoplasm for the cells in the entire selected area were calculated and documented for each tumor.

Clinicopathological patient follow-up data including information regarding potential metastasis and cause of death was collected from patient charts and treatment registers (Table 3).

Data was also gathered from a second cohort comprised of 80 uveal melanoma patients from TCGA via the National Cancer Institute, National Institutes of Health, USA (Table 1). The 2017 publication by Robertson et al. provided anonymized patient and tumor information, including whole exome sequencing results, in the Appendix A [50]. This information was downloaded to evaluate RNA sequencing data, measured in transcripts per million (TPM), for the receptors MTNR1A, MTNR1B, RORα, GPR50, and NQO2, and its relation to UM-related death and prognostic factors including tumor cell type, BAP1 mutation, and monosomy 3. Clinical information as well as pathology data regarding these prognostic factors were also obtained from TCGA for all 80 patients.

### 3.2. Statistical Methods

To compare potential differences in receptor expression as it correlated to uveal melanoma related death, *BAP1* mutation, monosomy 3 as well as tumor cell type, two-tailed Mann-Whitney U tests were employed using GraphPad Prism, version 10.1.1 (GraphPad Software, Boston, MA, USA). When making multiple comparisons, in the case of RNA expression levels across three tumor cell types, i.e., epithelioid, spindle and mixed, the Kruskal-Wallis test was employed. For the TCGA cohort, the sample size was 80 patients for analysis of all five receptors and for the St. Erik cohort 46 and 45 samples were analyzed for MTNR1A and MTNR1B respectively. Kaplan–Meier survival probability curves were plotted for patients when applicable using R, version 4.3.2 (R Core Team, Vienna, Austria), including the survival, survminer, ggplot2 and extrafont packages. For the St. Erik cohort, Kaplan–Meier curves were plotted for MTNR1A expression levels in the nucleus and cytoplasm of 46 tumor samples with a cut off below or above the median OD value. Similarly, for MTNR1B, Kaplan–Meier curves were plotted for expression levels in the nucleus and cytoplasm of 45 tumor samples. The difference in mean MTNR1A OD and MTNR1B OD in the nuclei and cytoplasm of the stained tumor tissue was determined for patients who died a UM related death and those who did not. To evaluate expression level significance, tumors with a mean MTNR1A OD or MTNR1B OD above the median value were compared with tumors with a mean MTNR1A OD or MTNR1B OD equal to or smaller than the median value.

Tumors from the TCGA cohort (*n* = 80) were considered to express the respective receptors if the mean RNA expression value was above 1 TPM (transcript per million). Note, the cutoff of 1 TPM for expression was arbitrarily chosen, as used in previous research [50]. No analyses in relation to the 1 TPM cutoff were conducted for MTNR1B RNA and GPR50 RNA considering that all tumors had TPM of less than 1. Similarly, the 1 TPM cutoff was not used for NQO2 as all tumors had TPM of more than 1 and analysis comparing groups above or below the cut off of 1 TPM would not be possible. Median TPM values were used as a cutoff in additional analyses for expression levels of MTNR1A RNA, NQO2 RNA, and RORα RNA obtained via the TCGA. This was not done for MTNR1B RNA or GPR50 RNA as they had a median value of 0.

Differences were considered significant when *p* < 0.05 and corrected *p* values were calculated using the Holm-Bonferroni method to limit error due to multiple comparisons for each cohort and prognostic factor. In other words, in the TCGA cohort (*n* = 80), the Holm-Bonferroni method was used to compare the five separate *p* values obtained for MTNR1A, MTNR1B, GPR50, RORα for each prognostic factor (i.e., epithelioid or non-epithelioid cell type, Monosomy 3 or Disomy 3, and *BAP1* mutation or wildtype. In the St. Erik cohort, the Holm-Bonferroni correction was used for the *p* values obtained for MTNR1A OD (*n* = 46) and MTNR1B OD (*n* = 45) in the cytoplasm and nucleus as they related to patients who did or did not experience a UM-related death. Note that, in the results tables, *p* values greater than or equal to 0.05 are labeled non-significant (ns), *p* values less than 0.05 are labeled with *, *p* values less than 0.01 are labeled with **, and *p* values less than 0.001 are labeled with ***.

## 4. Discussion

### 4.1. Main Findings

The objective of this research was to assess the expression levels of melatonin receptors in primary uveal melanoma and explore potential associations between receptor expression and the likelihood of uveal melanoma-related mortality. A key finding of this study is the presence of MTNR1A expression in uveal melanoma tumors in both the St. Erik and the TCGA cohorts. This, to the knowledge of the authors, is the first investigation of MTNR1A in human uveal melanoma tissue. As mentioned, one previous study has found the expression of MTNR1B in a uveal melanoma cell line [30]. The current study confirms this earlier finding. In the St. Erik cohort, these main melatonin receptors were minimally expressed, however, the mean value for staining intensity in the cytoplasm was significantly higher than in the nuclei (*p* < 0.0001) for both MTNR1A OD and MTNR1B OD. This likely corresponds to the fact that both receptors are transmembrane receptors and therefore positioned closer to the cytoplasm compared to the nucleus [51].

MTNR1A OD levels in the cytoplasm of UM cells were significantly higher in patients who died of UM compared to those who did not. Other studies investigating immunohistochemically stained tissues from other cancer types have found similar results with higher levels of MTNR1A in the tumors of patients with more advanced stages and worse prognosis [52,53]. One hypothesis for this observation, mentioned by Wang et al., is that melatonin levels appear to be lower in cancer patients compared to healthy individuals and even lower in those with more advanced cancer stages. This may in turn trigger a feedback mechanism where melatonin receptor expression is upregulated leading to increased levels of MTNR1A [52].

Nevertheless, our results showed no difference in receptor expression levels in more aggressive tumors compared to less aggressive tumors, as illustrated by the Kaplan–Meier analyses for both the St. Erik and TCGA cohorts. However, in our study, expression levels were based on IHC staining intensities for the St. Erik cohort and RNA sequencing data for the TCGA cohort, therefore, these results should be confirmed in future studies which look at protein expression specifically. Moreover, as the overall expression of MTNR1A in UM cells was relatively low, it is difficult to draw specific conclusions regarding this finding. Furthermore, tumors with the *BAP1* mutations in the TCGA cohort had significantly lower MTNR1A RNA expression levels (Holm–Bonferroni corrected *p* = 0.005). This is, however, a preliminary result which should be investigated further.

The current study also confirms the presence of RORα and NQO2 in uveal melanoma cells. While little is known about the physiologic effect of these receptors in uveal melanocyte homeostasis, their very presence in melanoma cells leaves room for potential involvement in the anti-cancer mechanisms of melatonin. NQO2 has been discussed in various models for cancer pathogenesis, and thus, has been considered a potential anti-cancer drug target [34]. Our study found NQO2 expression to be higher in patients with epithelioid tumors (*p* = 0.01), though the Holm–Bonferroni corrected *p* value was not significant (*p* = 0.05). Some previous studies have shown that higher levels of NQO2 may be associated with a poorer prognosis in non-small cell lung cancer, where the levels seem to be even higher in patients at a later stage of disease [34,54]. Since the epithelioid cell type is associated with worse prognosis, increased NQO2 expression in this group seems plausible. This is complicated by the ambivalent nature of NQO2 described in the literature. While earlier research highlights NQO2 as a detoxifying molecule, newer findings imply NQO2 is a mediator of increased ROS production and cytotoxicity. It is possible that the degree to which the molecule exerts either role is dependent on tumor stage and polymorphisms [55]. Melatonin was previously demonstrated to inhibit NQO2 at pharmacological concentrations [56]. While this study did not find a difference in survival between NQO2 expressing and non-expressing tumors, it is possible to hypothesize that the inhibitory effect is absent in uveal melanoma at endogenous melatonin concentrations. Future studies in patients using melatonin as an adjuvant treatment might shed light on this theory.

As for RORα, while not a receptor for melatonin, it has been shown that melatonin treatment correlates with increased RORα expression [44]. While survival was independent of RORα expression in the current study, RORα-related anti-tumor mechanisms could potentially be amplified via melatonin at therapeutic doses.

Lastly, while not a direct receptor for melatonin, earlier research suggests that GPR50 modulates the function of MTNR1A, which in turn inhibits the response of MTNR1A to binding by melatonin [46]. No tumors expressed GPR50 in this study, however, considering the potential role of MTNR1A in tumor suppression based on previous research, future studies investigating GPR50 may be of value.

### 4.2. Context

As previously stated, an earlier study found that MTNR1A and MTNR1B agonists as well as melatonin had a beneficial effect on uveal melanoma cells [30]. We have shown the presence of melatonin receptor expression in uveal melanoma, offering a potential mechanism for melatonin’s oncostatic properties. While this study did not show a clear correlation between melatonin receptor expression levels and survival in uveal melanoma patients, future studies investigating protein expression specifically should be performed to confirm these results. It is possible that melatonin exerts its potential anticancer effects via (1) its receptors at pharmacological doses rather than physiological doses, and (2) mechanisms independent of direct receptor-mediated signaling in tumor cells. One aspect to consider in future studies is the impact of varying pigmentation levels of melanin as it relates to melatonin receptor expression, considering the potential of pigmentation levels as an additional prognostic factor for UM [57].

### 4.3. Strengths and Limitations

This research involved data collection from two retrospective cohorts, which could potentially introduce biases associated with patient selection as well as the accuracy of the data. This restricts our capacity to draw conclusions regarding causal relationships. Selection bias could be present as the patients included in this study have undergone enucleation, the primary treatment option for larger and more advanced tumors, therefore, they may have a more aggressive variant compared to the general population of patients with UM. Another limitation is that this study did not examine aryl hydrocarbon receptor (AhR) or peroxisome proliferator-activated receptor gamma (PPARg), two newly identified nuclear melatonin receptors [58]. Additionally, this study did not examine all potential tumor-related effects of melatonin, as it is a molecule with many binding sites and diverse impacts. Furthermore, the phenotypic effects of melatonin are not necessarily mediated by melatonin itself but can also be mediated by its downstream metabolites [59]. One should also take into consideration the inherent limitations of immunohistochemistry which was used to analyze the uveal melanoma tissue from the St. Erik cohort. While IHC is a commonly used technique worldwide, the method is restricted concerning reproducibility, sensitivity, and specificity. Furthermore, receptor expression in the TCGA cohort was analyzed using RNA sequencing data measured in TPM. While TPM measurements provide insights into gene expression at the transcript level, they have limitations when inferring exact protein expression levels. It would, therefore, be of interest to confirm our results with future studies which address protein expression specifically. Lastly, while this study evaluates the expression of melatonin receptors in uveal melanoma tissue, it does not investigate their functional significance or downstream effects following activation or inhibition by melatonin. Additional studies are needed to determine potential biological implications of the observed receptor expression levels.

## 5. Conclusions

In conclusion, our findings confirm the presence of melatonin receptor expression in uveal melanoma tumors. Future studies should confirm the protein expression of these receptors, which may provide sites for melatonin’s binding or indirect impact.

## Figures and Tables

**Figure 1 ijms-25-08711-f001:**
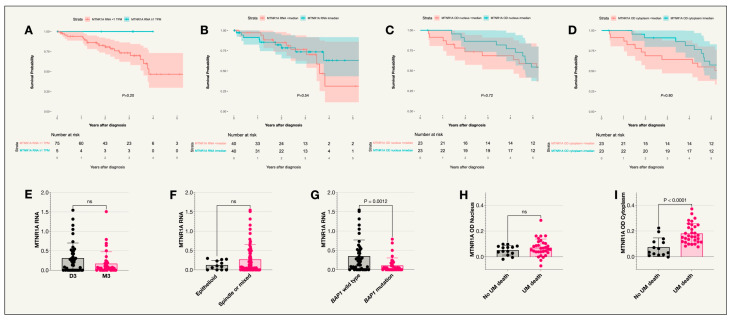
Kaplan–Meier survival curves and scatter plots for MTNR1A expression. (**A**) Survival curve comparing incidence of UM-related death in patients from the TCGA cohort with an MTNR1A RNA expression of less than 1 TPM compared to 1 TPM or above (*p* = 0.20), *n* = 80. (**B**) Survival curve comparing incidence of UM-related death in patients from the TCGA cohort with an MTNR1A RNA expression less than or equal to and above the median value (*p* = 0.54), *n* = 80. (**C**) Survival curve comparing the incidence of UM-related death in patients with IHC staining optical densities below or equal to and above the median in the nucleus (*p* = 0.72), *n* = 46. (**D**) Survival curve comparing the incidence of UM-related death in patients with IHC staining optical densities below or equal to and above the median in the cytoplasm (*p* = 0.80), *n* = 46. (**E**) Scatter plot comparing MTNR1A RNA expression in patient tumors with disomy 3 or monosomy 3. (**F**) Scatter plot comparing MTNR1A RNA expression in patients with tumors of an epithelioid cell type compared to other cell types (spindle or mixed). (**G**) Scatter plot comparing MTNR1A RNA expression in patients with the BAP1 wild type or BAP1 mutation where levels were higher in patients with the BAP1 wildtype (*p* = 0.0012). (**H**) Scatter plot comparing MTNR1A OD in the nucleus of tumor cells for patients who did or did not die due to uveal melanoma. (**I**) Scatter plot comparing MTNR1A OD in the cytoplasm of tumor cells for patients who did or did not die due to uveal melanoma (*p* < 0.0001). Colored fields on the Kaplan–Meier curves indicate 95% confidence intervals. ns = not significant. TCGA cohort, St Erik cohort.

**Figure 2 ijms-25-08711-f002:**
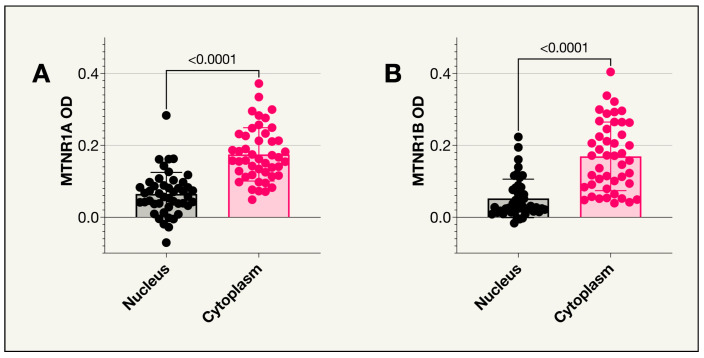
MTNR1A and MTNR1B receptors in the nucleus vs. cytoplasm. (**A**) The mean optical density (OD) for MTNR1A was significantly higher in the cytoplasm compared to the nucleus (*p* < 0.0001, *n* = 46); (**B**) The mean optical density (OD) for MTNR1B was significantly higher in the cytoplasm compared to the nucleus (*p* < 0.0001, *n* = 45).

**Figure 3 ijms-25-08711-f003:**
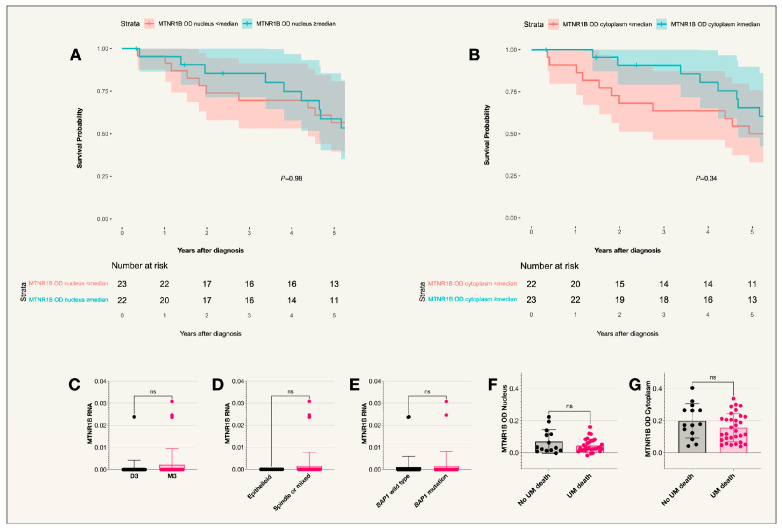
Kaplan–Meier survival curves and scatter plots for MTNR1B expression. (**A**) Survival curve comparing the incidence of UM-related death in those with MTNR1B OD levels in the nucleus below or equal to and above the median value (*p* = 0.98), *n* = 45. (**B**) Survival curve comparing incidence of UM-related death in those with MTNR1B OD levels in the cytoplasm below or equal to and above the median value (*p* = 0.34), *n* = 45. (**C**) Scatter plot comparing MTNR1B RNA expression in patient tumors with disomy 3 or monosomy 3. (**D**) Scatter plot comparing MTNR1B RNA expression in patients with tumors of an epithelioid cell type compared to other cell types (spindle or mixed). (**E**) Scatter plot comparing MTNR1B RNA expression in patients with the BAP1 wild type or BAP1 mutation. (**F**) Scatter plot comparing MTNR1B OD levels in the nucleus of tumor cells for patients who did or did not die due to uveal melanoma. (**G**) Scatter plot comparing MTNR1B OD levels in the cytoplasm of tumor cells for patients who did or did not die due to uveal melanoma. Colored fields on the Kaplan–Meier curves indicate 95% confidence intervals. ns = not significant. TCGA cohort, St Erik cohort.

**Figure 4 ijms-25-08711-f004:**
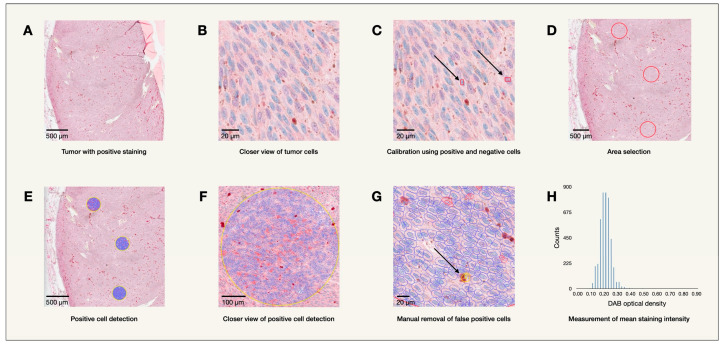
IHC data analysis. Using QuPath Bioimage analysis v. 0.4.1, mean DAB (3,3′-diaminobenzidine) staining intensity, measured by optical density (OD) for melatonin receptors were calculated from immunohistochemically stained primary uveal melanoma tissue. (**A**) Example of uveal melanoma tissue with positive staining where the presence of DAB (purple-red in color) signifies the presence of melatonin receptors. Cells which appear blue due to background hematoxylin staining, lack the presence of DAB and are therefore considered negative. (**B**) A closer view of stained tissue where positive (purple) and negative (blue) cells are visualized. (**C**) Calibration is performed by manually selecting a positive and negative cell. Note, the arrow to the left indicates a negative cell while the arrow to the right indicates a positive cell. (**D**) Three circular sections 500 μm in diameter are selected for analysis. (**E**) Positive Cell detection is performed to identify positive and negative cells within the selected sections. (**F**) A closer view of the detection of positive cells by QuPath. (**G**) An example of a false positive cell, indicated by an arrow, which is removed manually. (**H**) An example of the mean staining intensity for the entire selected area.

**Table 1 ijms-25-08711-t001:** Mean DAB Optical Density (OD) for UM-related death or non-UM related death.

Analysis	UM-Related Death, Mean DAB OD (SD)	Non UM-Related Death, Mean DAB OD (SD)	*p* (Holm–Bonferroni Corrected *p* Value)
MTNR1A OD in Nucleus and UM death	0.07 (0.07)	0.05 (0.04)	ns (ns)
MTNR1A OD in Cytoplasm and UM death	0.18 (0.08)	0.16 (0.07)	*** (**)
MTNR1B OD in Nucleus and UM death	0.04 (0.04)	0.07 (0.07)	ns (ns)
MTNR1B OD in Cytoplasm and UM death	0.16 (0.09)	0.20 (0.11)	ns (ns)

*p* values less than 0.01 are labeled with **, and *p* values less than 0.001 are labeled with ***. ns = not significant.

**Table 2 ijms-25-08711-t002:** (**A**) Mean RNA expression across cytomorphology. (**B**) Mean RNA expression for monosomy 3 vs. disomy 3. (**C**) Mean RNA expression for *BAP1* mutation or wildtype.

(A)			
Analysis	Epithelioid, Mean TPM (SD)	Non-Epithelioid, Mean TPM (SD)	*p* (Holm-Bonferroni Corrected *p* Value)
NQO2 vs. epithelioid or non-epithelioid	50 (17)	35 (16)	* (ns)
RORA vs. epithelioid or non-epithelioid	1 (0.8)	0.9 (1)	ns (ns)
GPR50 vs. epithelioid or non-epithelioid	<0.05 (<0.05)	<0.05 (<0.05)	ns (ns)
MTNR1B vs. epithelioid or non-epithelioid	0 (0)	<0.05 (<0.05)	ns (ns)
MTNR1A vs. epithelioid or non-epithelioid	0.1 (0.1)	0.3 (0.4)	ns (ns)
(**B**)			
**Analysis**	**Monosomy 3,** **Mean TPM (SD)**	**Disomy 3,** **Mean TPM (SD)**	***p* (Holm** **–Bonferroni Corrected *p* Value)**
NQO2 vs. M3 or D3	41 (18)	36 (15)	ns (ns)
RORA vs. M3 or D3	1 (1)	0.9 (1)	ns (ns)
GPR50 vs. M3 or D3	<0.05 (<0.05)	<0.05 (<0.05)	ns (ns)
MTNR1B vs. M3 or D3	<0.05 (<0.05)	0 (<0.05)	ns (ns)
MTNR1A vs. M3 or D3	0.4 (0.3)	0.3 (0.2)	ns (ns)
(**C**)			
**Analysis**	***BAP1* Mutation,** **Mean TPM (SD)**	***BAP1* Wildtype,** **Mean TPM (SD)**	***p* (Holm** **–Bonferroni Corrected *p* Value)**
NQO2 vs. BAP1 mutation or wildtype	41 (17)	36 (16)	ns (ns)
RORA vs. BAP1 mutation or wildtype	1.1 (1.1)	0.9 (0.9)	ns (ns)
GPR50 vs. BAP1 mutation or wildtype	<0.05 (<0.05)	<0.05 (<0.05)	ns (ns)
MTNR1B vs. BAP1 mutation or wildtype	<0.05 (<0.05)	<0.05 (<0.05)	ns (ns)
MTNR1A vs. BAP1 mutation or wildtype	0.1 (0.2)	0.4 (0.4)	*** (***)

ns = not significant. *p* values less than 0.05 are labeled with *, and *p* values less than 0.001 are labeled with ***.

**Table 3 ijms-25-08711-t003:** Clinicopathologic characteristics of the two included cohorts.

	St. Erik Eye Hospital Cohort	TCGA Cohort
*n*	47	80
Sex, *n* (%)		
Female	19 (40)	35 (44)
Male	28 (60)	45 (56)
Ciliary body involvement, *n* (%)	8 (16)	10 (13)
Extraocular extension, *n* (%)	7 (14)	6 (8)
Tumor thickness at diagnosis, mean mm (SD)	7.8 (3.8)	10.8 (2.6)
Tumor diameter at diagnosis, mean mm (SD)	14.7 (4.9)	16.6 (3.8)
Cell type, *n* (%)		
Spindle	18 (38)	28 (35)
Epithelioid	11 (24)	12 (15)
Mixed Spindle and Epithelioid	18 (38)	39 (49)
Not available	0 (0)	1 (1)
AJCC T-category, *n* (%)		
T1a	5 (11)	0 (0)
T1b	0 (0)	0 (0)
T1c	2 (4)	0 (0)
T2a	9 (19)	12 (15)
T2b	1 (2)	2 (3)
T2c	1 (2)	0 (0)
T3a	11 (24)	26 (32)
T3b	1 (2)	5 (6)
T3c	3 (6)	1 (1)
T4a	3 (6)	20 (25)
T4b	4 (9)	9 (11)
T4c	7 (15)	2 (3)
T4d	0 (0)	2 (3)
T4e	0 (0)	1 (1)
AJCC stage at diagnosis, *n* (%)		
I	5 (11)	0 (0)
IIA	11 (23)	4 (5)
IIB	12 (26)	32 (40)
IIIA	9 (19)	27 (34)
IIIB	10 (21)	10 (12)
IIIC	0 (0)	3 (4)
IV	0 (0)	4 (5)
Primary treatment, *n* (%)		
Plaque brachytherapy	17 (36)	40 (50)
Enucleation	30 (64)	40 (50)
Median follow-up, years (IQR)		
Death from metastatic uveal melanoma	4.6 (3.4)	1.7 (1.6)
Death from other cause	9.7 (4.8)	1.6 (1.8)
Alive	-	2.3 (1.6)
AJCC, American Joint Committee on Cancer. IQR, Interquartile range. SD, Standard deviation.		

## Data Availability

The raw, anonymized data from the St. Erik cohort supporting the conclusions of this article will be made available by the authors on request. Data from the TCGA cohort are openly available at https://www.cancer.gov/tcga, accessed on 4 May 2023.

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
