# Peer review of "Melatonin Receptor Expression in Primary Uveal Melanoma"

_ijms, 2024, doi:10.3390/ijms25168711_

Round 1

Reviewer 1 Report

Comments and Suggestions for Authors

General comments:

I)           The eventual correlation between melatonin receptor expression in primary uveal melanoma (UM) and UM-related mortality and prognostic was assessed and no correlations were observed. the expressions of RORα, NQO2, or GPR50 were also not correlated with survival probability or prognostic factors. The novelty of the findings were incremental and not well supported by experimental evidence. 

II)          Materials and methods did not contain sufficient information on how experiments were conducted. It was not possible to assess the reliability of the findings. The statistical analysis was unclear and lacked details.

III)            Gene expressions were not confirmed by their respective gene products to support adequately the findings. Since there were no correlation between gene expressions and UM-mortality, their respective protein expressions could not be affected. 

IV)            The findings associated to BAP1 mutations in the TCGA cohort with lower MTNR1A RNA expression levels need to be confirmed by further experiments. 

V)             It was unclear why the expression levels of RORα, NQO2, or GPR50 were investigated, especially that their possible associations with melatonin receptors were not clearly justified. 

VI)            As the authors pointed out, the present work had several limitations, including data collection with two retroperspective cohorts, section bias, and limitations of immunohistochemistry. 

Minor comments:

1)    Abstract, p1: Delete “Uveal melanoma (UM) poses a significant clinical challenge due to its high metastasis rate and poor prognosis.”

2)    Abstract, p1: Specify MTNR abbreviations in “Immunohistochemical analysis of 47 primary UM tissues showed low expression of MTNR1A and MTNR1B receptors, with MTNR1A significantly higher in patients who succumbed UM.

3)    Abstract, p1: Specify RORα, NQO2, and GPR50 abbreviations in “Analysis of TCGA data from UM patients revealed RNA expression for MTNR1A, RORα, and NQO2, but not MTNR1B or GPR50.

4)    Abstract, p1: Specify BAP1 abbreviation in “Higher MTNR1A RNA levels were observed in patients with a BAP1 mutation, and higher NQO2 RNA levels were noted in patients with the epithelioid tumor cell type.

5)    Abstract, p1: Delete “Further research is needed to explore additional mechanisms of melatonin’s oncostatic effects, including its antioxidant properties and immune modulation.

6)    Introduction, p1: Specify BRAF abbreviation in “Unlike cutaneous melanoma, treatments involving immune checkpoint and BRAF inhibition have shown limited efficacy in extending survival for those with metastasized uveal melanoma [7].

7)    Introduction, p1: Delete “There is therefore a need for improved treatment options which lower the risk for systemic disease.

8)    Introduction, p1: Justify, and add references to support “Since macrometastases are uncommon at diagnosis, predicting the prognosis for individual patients relies on clinical and histological aspects of the primary tumor.

9)    Introduction, p1-2: Specify BRCA-1 abbreviation in BRCA-1 melanomas consist of four cell types; normal; spindle A; spindle B; and epithelioid cells 43 where the last-mentioned has lower expression levels of the BRCA-1 associated protein (BAP-1)[9].

10) Introduction, p2: Add references to support “When used as an adjuvant treatment for various types of cancer in previous studies, melatonin has been associated with inhibited tumor growth, increased one-year survival, and few side effects.

11) Introduction, p2: Delete “In regard to the potential therapeutic effect in uveal melanoma, however, research is limited.

12) Materials and Methods, Aims of the study, p2: Shorten and merge into the last part of the introduction: “This study aimed to explore expression levels for melatonin receptors in primary uveal melanoma and investigate a potential correlation between receptor expression and the risk for uveal melanoma related mortality as well as other prognostic factors such as BAP1 expression, cell type, and monosomy 3. If melatonin influences the progression of uveal melanoma and slow the onset of macrometastases, one potential mechanism could be through binding melatonin receptors present in primary uveal melanoma cells.

13) Materials and Methods, Patients and Samples, 2: Update ethical information by addition of an appendix record concerning the 2000-2008 collected eyes due to inconsistent date of approval (Reference number 2024-00295-02) in “48 enucleated eyes containing primary uveal melanoma tumors from the years 2000-2008 were collected from the archives of the St. Erik Ophthalmic Pathology Laboratory for this study.”

14) Materials and Methods, Patients and Samples, 3: Specify types of monoclonal antibodies and add more information on the staining to support “Formalin fixed paraffin embedded tissue from the enucleated eyes were sectioned and stained on a BOND III IHC stainer by biomedical analysts using mouse monoclonal antibodies.”

15) Materials and Methods, Patients and Samples, 3: Replace “optical staining density” by “absorbance ” in “Mean levels of immunohistochemical expression of MTNR1A and MTNR1B across all tumor cells, as determined by their optical staining density in bioimage analysis (MTNR1A OD, MTNR1B OD), were analyzed using the program QuPath Bioimage analysis v. 0.4.1 m4 (Bankhead et al., 2017) as illustrated in Figure 1.”

16) Materials and Methods, Patients and Samples, Figure 1, p2: Move Figure 1 and add comments to support findings from figure 1in the result section. 

17) Materials and Methods, Patients and Samples, Figure-1 legend, p3: Replace “optical densities” by “absorbance” and specify DAB abbreviation in “Figure 1. IHC data analysis. Using QuPath Bioimage analysis v. 0.4.1, mean DAB optical densities for melatonin receptors were calculated from immunohistochemically stained primary uveal melanoma tissue.”

18) Materials and Methods, Patients and Samples, Figure 1, p3: Indicate Table 1 in the text, and eventually reference since it was a previous clinical register (2000-2008) to support Clinicopathological patient follow-up data including information regarding potential metastasis and

19) Materials and Methods, Melatonin receptors MTNR1A (MT1) and MTNR1B (MT2), Title, p5: Delete abbreviations (MT1) and (MT2) in the title.

20) Materials and Methods, Melatonin receptors MTNR1A (MT1) and MTNR1B (MT2), p5-6: Materials and Methods is not an introduction. Merge in the introduction “Melatonin impacts various processes in the human body primarily through two membrane receptors; melatonin receptor type 1A (MTNR1A) and melatonin receptor type 1B (MTNR1B) [28]. Both of these receptors are G-coupled receptor proteins widely expressed both centrally and peripherally throughout the human body and play a role in melatonin’s impact on physiological systems such as circadian rhythm, neurodevelopment, blood glucose regulation and the cardiovascular system [29,30]. In a study published in 2000, Roberts et al identified the MTNR1B subtype, but not the MTNR1A subtype, in reverse-transcribed RNA obtained from normal uveal melanocytes as well as melanoma cell lines [31]. In the study, receptor agonists for both MTNR1A and MTNR1B as well as melatonin itself inhibited the growth of uveal melanoma cells at physiological concentrations, thereby suggesting a receptor-mediated mechanism for the inhibition of tumor cell growth [31]. MTNR1B appeared to be expressed to a larger extent in the cancerous cell lines compared to normal uveal melanocytes, though this observation was not specifically verified [31].”

21) Materials and Methods, Melatonin receptors MTNR1A (MT1) and MTNR1B (MT2), p5: Explain how MTNR1A and MTR1B were determined and in which tissues.

22) Materials and Methods, NQO2, Title, p6: Replace NQO2 by its full name in the title.

23) Materials and Methods, NQO2, p6: Delete or merge into the introduction “N-ribosyldihydronicotinamide:quinone oxidoreductase 2 (NQO2) presents a melatonin binding site known as MT3 and carries out two-electron reductions of primarily quinones [32,33]. Despite previously being considered a detoxifying enzyme, some studies suggest that NQO2 activation leads to the production of reactive oxygen species (ROS) and the enzyme has been found to be increased in some cancer cell lines [34,35]. Of note, the affinity of NQO2 to melatonin seems to be in the nanomolar range, however, melatonin appears to inhibit the enzyme within the micromolar range, where concentrations above 1 μM are considered pharmacological [33].”

24) Materials and Methods, NQO2, p6: Explain how NQO2 was determined and in which tissues.

25) Materials and Methods, GPR50, Title, p6: Replace GPR50 by its full name in the title.

26) Materials and Methods, GPR50, p6: Delete or Merge into the introduction “GPR50 is a G protein-coupled receptor with the ability to heterodimerize with both MTNR1A and MTNR1B [36]. GPR50 does not seem to modify the function of MTNR1A but rather leads to an inhibition of the functional response of the receptor to stimulation by melatonin [36].

27) Materials and Methods, GPR50, p6: Explain how GPR50 was monitored and in which tissues.

28) Materials and Methods, RORα, Title, p6: Replace “RORα” by its full name in the Title.

29) Materials and Methods, RORα , p6: Delete or merge in the introduction “The retinoic acid-related orphan receptor alpha (RORα) is a nuclear receptor within the ROR family [37]. RORα has been found to play a role in the immune system by contributing to Th17 development and the generation of innate lymphoid cells [38,39]. It may also aid in the stabilization and transcription of p53 [40]. Several studies suggest that RORα expression is down-regulated during tumor development and progression, while exogenous RORα inhibits cell proliferation and tumor growth in colorectal, prostate and breast cancer [41–43]. Whether a correlation between melatonin and the ROR family exists is still debated [44]. Earlier studies suggested that melatonin acts as a ligand to RORα, though a later study found this does not occur in a high-affinity manner [45,46]. One study, using crystallography and molecular modeling, indicated that melatonin is unlikely to function as an ROR ligand [47]. Despite this, specific intermediate steps enabling melatonin to indirectly regulate ROR expression and function have been confirmed [44].”

30) Materials and Methods, RORα , p6: Explain how RORα was determined and in which tissues. 

31) Materials and Methods, p6: Indicate how bio-informatic analysis (including Kaplan-Meier analysis) was performed.

32) Materials and Methods, Statistical methods, p6:Indicate numver of samples to support “To compare potential differences in receptor expression as it correlated to uveal melanoma related death, BAP1 mutation, monosomy 3 as well as tumor cell type, two-tailed Mann-Whitney U tests were employed using GraphPad Prism, version 10.1.1 (GraphPad Software, Boston, Massachusetts USA).

33) Materials and Methods, Statistical methods, p6: Add further information in the Materials and methods, bio-informatic section and indicate number of patients to support “Kaplan-Meier survival probability curves were plotted for patients with expression levels below or above the median value, as well as for those with expression levels below or above 1 TPM (transcript per million), using R, version 4.3.2 (R Core Team, Vienna, Austria), including the survival, survminer, ggplot2 and extrafont packages.

34) Materials and Methods, Statistical methods, p6: There was no information in materials and methods how cells were extracted and from which tissues. Rephrase “The difference in mean MTNR1A OD and MTNR1B OD in the nuclei and cytoplasm of tumor cells was determined for patients who died a UM related death and those who did not.”

35) Materials and Methods, Statistical methods, p6: It was unclear. Add further information to support “For the Kaplan-Meier analysis, tumors with a mean MTNR1A OD or MTNR1B OD above the median value were compared with tumors with a mean MTNR1A OD or MTNR1B OD equal to or smaller than the median value.

36) Materials and Methods, Statistical methods, p6: Specify TPM abbreviation and add further information on TCGA cohort to support “Tumors from the TCGA cohort were considered to express the respective receptors if the mean value was above 1 TPM.”

37) Materials and Methods, Statistical methods, p6: Justify “Similarly, the 1 TPM cutoff was not used for NQO2 as all tumors had TPM of more than 1.

38) Materials and Methods, Statistical methods, p6: There was no information how RNA analysis was performed and in which tissues to support “Median TPM values were used as a cutoff in additional analyses for MTNR1A RNA, NQO2 RNA, and RORα RNA.

39) Materials and Methods, Statistical methods, p6: Specify cohort, indicate number of samples, and merge in the appropriate paragraph where the Holm-Bonferroni method was used since that not all the statistical findings can be treated by the Holm-Bonferroni method to support “Differences were considered significant when P < 0.05 and corrected P values were calculated using the Holm-Bonferroni method to limit error due to multiple comparisons for each cohort and prognostic factor.”

40) Materials and Methods, Statistical methods, p6: Specify which values, and indicate number of independent measurements to support “Note that, in the results tables, P values greater than or equal to 0.05 are labeled non-significant (ns), P values less than 0.05 are labeled with *, P values less than 0.01 are labeled with **, and P values less than 0.001 are labeled with ***.”

41) Results, p7: Delete “This section may be divided by subheadings. It should provide a concise and precise description of the experimental results, their interpretation, as well as the experimental Conclusions that can be drawn.”

42) Results, Descriptive statistics, MTNR1A, Title, p7: Replace MTR1A by its full name.

43) Results, Descriptive statistics, MTNR1A, p7: Materials and methods did not contain any information how expression of MTNR1A was determined. Add the requested information in materials and methods to support “MTNR1A was expressed in uveal melanoma tumors in both the St. Erik and the TCGA cohorts (Figure 2).

44) Results, Descriptive statistics, MTNR1A, Figure 2, p7: Enlarge size of letters in Figure 2. “MTNR1A was expressed in uveal melanoma tumors in both the St. Erik and the TCGA cohorts (Figure 2).

45) Results, Descriptive statistics, MTNR1A, Figure-2 legend, p7: Apparently n=80 does not correspond to the number of eyes (n=46). Resolve the contradiction in “A. Survival curve comparing incidence of UM-related death in those with an MTNR1A RNA expression of less than 1 TPM compared to 1 TPM or above (P=0.20)*, n=80.”

46) Results, Descriptive statistics, MTNR1A, p7: It was unclear. Replace OD by full name in “Similarly, no significant difference was seen in survival probability between patients in the St. Erik cohort with MTNR1A OD values below vs above the median.

47) Results, Descriptive statistics, MTNR1A, p7: It was unclear. Delete “In the St. Erik cohort, slides were missing for one of the tumors resulting in a total of 46 tumors analyzed.

48) Results, Descriptive statistics, MTNR1A, p7: No information in Materials and methods concerned cellular findings. Add the requested information in Materials and methods to support “As illustrated in Figure 3, the median MTNR1A OD was significantly higher in the cytoplasm compared to the nuclei (P<0.001).

49) Results, Descriptive statistics, MTNR1A, Figure-3 legend, p7: Indicate number of independent measurements in the Figure-3 legend. 

50) Results, Descriptive statistics, MTNR1A, p7: Replace OD by full name in “No significant difference was seen between UM related death and nuclear MTNR1A OD.

51) Results, Descriptive statistics, MTNR1A, p7: Materials and methods did not contain any information on BAP1 mutations. Indicate the origin of the samples and how experiments were performed in Materials and methods to support “In the TCGA cohort, MTNR1A RNA levels were significantly lower in tumors with BAP1 mutations (Holm-Bonferroni corrected P=0.005) while there was no correlation between MTNR1A RNA and Monosomy 3 or cell type (Table 2 and Table 3).

52) Results, Descriptive statistics, MTNR1A, p7: There were several tables 2 and 3. Specify exactly which tables 2 and 3 to support “In the TCGA cohort, MTNR1A RNA levels were significantly lower in tumors with BAP1 mutations (Holm-Bonferroni corrected P=0.005) while there was no correlation between MTNR1A RNA and Monosomy 3 or cell type (Table 2 and Table 3).

53) Results, MTNR1B, Title, p8 : Replace MTNR1B by its full name in the title.

54) Results, MTNR1B, p8: it was unclear what were the missing slides. Rephrase “In the St. Erik cohort, expression levels were analyzed in the nuclei and cytoplasm of 45 tumors due to missing slides.”

55) Results, MTNR1B, p8: Replace OD by its full name in “As in the case of MTNR1A, the median MTNR1B OD was significantly higher in the cytoplasm compared to the nuclei (P<0.001).”

56) Results, MTNR1B, p8: Materials and methods did not contain any information how MTNR1B was determined in the cytoplasm and in the nuclei. Add the requested information in Material and Methods to support “As in the case of MTNR1A, the median MTNR1B OD was significantly higher in the cytoplasm compared to the nuclei (P<0.001).”

57) Results, MTNR1B, p8: It was unclear. Replace OD by its full name and specify what were the levels? in Similarly, no correlation was observed between UM related death and OD levels in the cytoplasm or nucleus (Figure 4, Table 2 and Table 3).

58) Results, MTNR1B, p8: Indicate which tables 2 to support Similarly, no correlation was observed between UM related death and OD levels in the cytoplasm or nucleus (Figure 4, Table 2 and Table 3).

59) Results, MTNR1B, Figure 4, p8: Enlarge size of letters in Figure 4 to support Similarly, no correlation was observed between UM related death and OD levels in the cytoplasm or nucleus (Figure 4, Table 2 and Table 3).

60) Results, MTNR1B, Figure 4, p8: Materials and methods did not contain any information on Kaplan-Meier survival curves and how it were determined and from which samples to support “Similarly, no correlation was observed between UM related death and OD levels in the cytoplasm or nucleus (Figure 4, Table 2 and Table 3).

61) Results, MTNR1B, Figure 4, p8: Number of samples (n=45) did not correspond to the number of eyes. Rephrase A. Survival curve comparing the incidence of UM-related death in those with MTNR1B OD levels in the nucleus below or equal to and above the median value (P=0.98)+, n=45.” And “B. Survival curve comparing incidence of UM-related death in those with MTNR1B OD levels in the cytoplasm below or equal to and above the median value (P=0.34)+, n=45.

62) Results, NQO2, Title, p11: Replace NQO2 by it full name in the title.

63) Results, NQO2, Title, p11: Materials and Methods lacked information on how was determined survival probability, and number of patients. Add the requested information in materials and methods. “There was no correlation between receptor expression and survival probability (Figure A1).”

64) Results, NQO2, Title, p11: Resolve contradiction in “Interestingly, NQO2 RNA expression was higher in patients with epithelioid tumors compared to those with either spindle or mixed cell types (P=0.01). However, after the Holm-Bonferroni correction, the result was no longer significant (corrected P=0.05).”

65) Results, NQO2, Title, p11: Materials and Methods lacked information on how was determined NQO2 RNA expression, from which samples, and number of samples. Add the requested information in materials and methods. “Interestingly, NQO2 RNA expression was higher in patients with epithelioid tumors compared to those with either spindle or mixed cell types (P=0.01). However, after the Holm-Bonferroni correction, the result was no longer significant (corrected P=0.05).”

66) Results, NQO2, Title, p11: Materials and Methods lacked information on Kruskal-Wallis test, which samples, and number of samples. Add the requested information in Materials and Methods. “Similarly, when comparing NQO2 RNA expression levels across the three separate cell types, i.e. epithelioid, spindle and mixed, using the Kruskal-Wallis test, the original P value was significant (P=0.0468) while the Holm-Bonferroni corrected P value was not (corrected P=0.234).”

67) Results, RORα, Title, p11 : Replace RORα by its full name in the title.

68) Results, NQO2, p11: It was unclear from which database originated the 80 patients? Add further information in Material and Methods to support “The median RNA expression level of RORα among the 80 tumors in the TCGA data was 0.66 TPM. 25 patients had an RNA expression level equal to or above 1 TPM while 55 patients had an expression level below 1 TPM. No correlation between expression and survival probability or prognostic factors was noted (Figure A2, Table 2).”

69) Results, NQO2, Figure A2 p11: Enlarge size of letters in Figure A2 to support “It was unclear from which database originated the 80 patients? Add further information in Material and Methods to support “The median RNA expression level of RORα among the 80 tumors in the TCGA data was 0.66 TPM. 25 patients had an RNA expression level equal to or above 1 TPM while 55 patients had an expression level below 1 TPM. No correlation between expression and survival probability or prognostic factors was noted (Figure A2, Table 2).”

70) Results, GPR50, Title, p11: Replace GPR50 by its full name in the Title.

71) Results, GPR50, p12: Figure 3A presented graphs on epithelioid cell type compared to other cell Types and on BAP1 wild type or BAP1 mutatio which were not commented to support No tumors from the TCGA data expressed GPR50 i.e. the median the expression level was 0 TPM with no tumors having a mean RNA expression level over 1 TPM (Figure A3)

72) Results, main findings, p12: Rephrase. It was unclear what refers OD in “The current study confirms this earlier finding. In the St. Erik cohort, these main melatonin receptors were minimally expressed, however, the mean value for expression in the cytoplasm was significantly higher than in the nuclei (P<0.0001) for both MTNR1A OD and MTNR1B OD.

73) Results, main findings, p12: A transmembrane receptor can’t be found in the cytoplasm. Rephrase “This likely corresponds to the fact that both receptors are transmembrane receptors and are therefore more likely to be present in the cytoplasm.”

74) Results, main findings, p12: This need to be confirmed by the expression of proteins to support “Nevertheless, there was no difference in receptor expression in more aggressive  tumors compared to less aggressive tumors as illustrated by the Kaplan-Meier analyses for both the St. Erik and TCGA cohort.

75) Results, context, p12:Alternatively the gene expression was not accompanied by the protein expression. Rephrase “While this study did not show a clear correlation between melatonin receptor expression and survival in uveal melanoma patients, the hormone itself may exert its potential anticancer effects via 1) its receptors at pharmacological doses rather than physiological doses 2) mechanisms independent of direct receptor mediated signaling in tumor cells.

76) Results, context, p12: It is speculative since it assumed that the protein expression increased with its gene expression. Delete “Different mechanisms of how this may be achieved have been proposed. Notably, melatonin has been implicated in altering the tumor microenvironment (TME) in different neoplasms [55]. TME is characterized by an interplay of immune cells, whereby cytotoxic T-lymphocytes (CTL) and natural killer (NK) cells are key mediators of tumor suppression. Other cell types like T regulatory cells (Tregs) or M2 macrophages limit the activity of CTLs and NK cells, thus limiting their anticancer activity [56]. Melatonin receptor mediated regulation of immune responses has been closely studied, and findings suggest that melatonin increases CTL and NK responses, while limiting Treg activity. Furthermore, the role of melatonin as an antioxidant and as a free radical scavenger has been studied extensively. While the amplitude of these effects in vivo is not known, current evidence largely supports that melatonin has the chemical properties of a scavenger molecule [57]. Thus, modifying TME immune responses and the chemical properties of melatonin in protection from ROS are promising branches via which melatonin can exert its anticancer effects beyond activation of its designated receptors.

77) Conclusions, p14: There were no findings associated to the expressions of proteins. Rephrase “In conclusion, our findings confirm the presence of melatonin receptors in uveal melanoma tumors, potentially providing sites for melatonin’s binding or indirect impact.

78) Conclusions, p14: There were no correlation between gene expression of receptor expression and uveal melanoma-related mortality Delete “While these receptors suggest a potential mechanism for melatonin's anti-cancer effects, our study did not establish a clear association between receptor expression and uveal melanoma-related mortality. Further investigation is warranted to explore alternative mechanisms through which melatonin may exert its anti-cancer properties in uveal melanoma, including its role as an antioxidant and free radical scavenger, as well as its potential for TME and immune system modulation.

Reviewer 2 Report

Comments and Suggestions for Authors

In this manuscript, authors have performed an evaluation of melatonin  receptor expression in primary UM and its association with UM-related mortality and prognostic  factors. They found from the IHC analysis that 47 primary UM tissues showed a low expression of  MTNR1A and MTNR1B receptors, with MTNR1A significantly higher in patients who succumbed  to UM. This is a nice work and manuscript is generally written well. However, I recommend authors to incorporate the following suggestions to enhance the quality and visibility of the paper. 

1. Can authors details a bit about the gender dependenices of their study results ?

2. How does the evaluation of melatonin  receptor expression varies on patients with different melanin pigmentation levels 

3. Do authors have any in vivo imaging results to show such as OCT, SLO fundus fluoroscence etc.

4. I urge authors to mention a brief sentence about the potential scope of investigating pigmentation changes in uveal melanoma tumors in the presence of melatonin receptors as future work with the suggested reference for pigmentary changes in ocular pathology[1-2]

(1) https://doi.org/10.1038/s41598-021-95320-z

(2) https://doi.org/10.1167/iovs.06-0122 

Round 2

Reviewer 1 Report

Comments and Suggestions for Authors

General comments:

I) The eventual correlation between melatonin receptor expression in primary uveal melanoma (UM) and UM-related mortality and prognostic was assessed and no correlations were observed. the expressions of RORα, NQO2, or GPR50 were also not correlated with survival probability or prognostic factors. The novelty of the findings were incremental and not well supported by experimental evidence.

Response: Thank you for your comment. Kaplan-Meier survival analyses for patients based on RORα, NQO2, or GPR50 expression are provided in the Appendix. Forgive us, but we do not fully understand the comment regarding our findings not being supported by experimental evidence? Does it intend to say that our conclusions were not supported by our result? If so, we respectfully disagree. Our conclusions were that 1) melatonin receptors are expressed in uveal melanoma, and that 2) future studies are recommendable are well supported by our results. These modest conclusions are well supported by the results of our study. 

Reviewer’s answer: It was insufficiently addressed. What was missing is further experimental evidence to adequately support the findings, more precisely the respective gene products were not determined and analyzed. That is why the findings appear preliminary to draw any firm conclusion. Besides the authors acknowledge that further experiments are needed to confirm the modest conclusion. 

II) Materials and methods did not contain sufficient information on how experiments were conducted. It was not possible to assess the reliability of the findings. The statistical analysis was unclear and lacked details

Response: Thank you for your comment. We have now revised the materials and methods section thoroughly, including the statistical methods section, which now give more detailed information regarding the immunohistochemical analysis of the tissue from the St. Erik Cohort, data collection from the TCGA, how the statistical analysis was performed as well as other points mentioned more specifically in the responses below. 

Reviewer’s answer: It was insufficiently addressed. It is now clear that it was essentially a statistical analysis based from previous records. There was no information, not even any references, which could support how experiments were performed. I can not assess the reliability of the findings. This is of crucial importance since it was unclear if there were any limitations due to the experimental data collected. 

III) Gene expressions were not confirmed by their respective gene products to support adequately the findings. Since there were no correlation between gene expressions and UM- mortality, their respective protein expressions could not be affected.

Response: Thank you for your comment. Unfortunately, protein expression levels were not available for the TCGA cohort, preventing its analysis. We agree that this is a limitation and have therefore included this in the limitations section: “Furthermore, receptor expression in the TCGA cohort was analyzed using RNA sequencing data measured in TPM. While TPM measurements provide insights into gene expression at the transcript level, they have limitations when inferring exact protein expression levels. It would therefore be of interest to confirm our results with future studies which address protein expression specifically.” 

Reviewer’s answer: It was not addressed. The authors acknowledged to confirm our results with future studies which address protein expression specifically.

VI) As the authors pointed out, the present work had several limitations, including data  collection with two retroperspective cohorts, section bias, and limitations of immunohistochemistry.

Response: Thank you, we agree fully. 

Reviewer’s answer: The authors acknowledge the limitations of the work. 
